Exergames for women with fibromyalgia: a randomised controlled trial to evaluate the effects on mobility skills, balance and fear of falling

Collado-Mateo Daniel dcolladom@unex.es 1
Dominguez-Muñoz Francisco J. 1
Adsuar Jose C. 1
Merellano-Navarro Eugenio 2
Gusi Narcis 1
1 Faculty of Sport Science, Departamento de Didáctica de la Expresión Musical, Plástica y Corporal, University of Extremadura , Cáceres , Spain
2 Facultad de Educación, Universidad Autónoma de Chile , Talca , Chile
Ramírez-Campillo Rodrigo
Electronic publication date: 2017 Apr 20
Publication date: 2017
Volume: 5
Electronic Location ID: e3211
Received 2016 Dec 20; Accepted 2017 Mar 21
Copyright: ©2017 Collado-Mateo et al.
Copyright year: 2017
Copyright holder: Collado-Mateo et al.
License: This is an open access article distributed under the terms of the Creative Commons Attribution License, which permits unrestricted use, distribution, reproduction and adaptation in any medium and for any purpose provided that it is properly attributed. For attribution, the original author(s), title, publication source (PeerJ) and either DOI or URL of the article must be cited.
License URL: https://creativecommons.org/licenses/by/4.0/

Keywords: Falls, Virtual reality, Chronic pain, Postural control

Funding: Spanish Ministry of Economy and Competitiveness DEP2012-39828 DEP2015-70356 Government of Extremadura and European Union Regional Development Funds (FEDER) GR10127 Spanish Ministry of Education, Culture and Sport FPU14/01283 In the framework of Spanish National R+D+i Plan, the current study has been co-funded by the Spanish Ministry of Economy and Competitiveness with the reference DEP2012-39828 and DEP2015-70356. This study has been also co-funded by the Government of Extremadura and European Union Regional Development Funds (FEDER), a way of making Europe (ref. GR10127). The author DCM is supported by a grant from the Spanish Ministry of Education, Culture and Sport (FPU14/01283). The funders had no role in study design, data collection and analysis, decision to publish, or preparation of the manuscript.

==============================
Background

Exergames are a new form of rehabilitation that combine the characteristics of physical exercise and the benefits of non-immersive virtual reality (VR). Effects of this novel therapy in women fibromyalgia are still unknown. The objective was to evaluate the effects of exergame-based intervention on mobility skills, balance and fear of falling in women with fibromyalgia.

Methods

This study was a randomized controlled trial with concealed allocation. Seventy-six women with fibromyalgia were divided into two groups: the exercise group received an eight week intervention based on exergames, while the control group continued their usual activities. Mobility skills were evaluated using the timed up and go test, while balance was assessed using the functional reach test, and the CTSIB protocol. Fear of falling was evaluated on a scale of 0–100 (0, no fear; 100, extreme fear). Measurements were performed before and after the intervention. A repeated-measures linear mixed model was used to compare the effects of the intervention between the two groups.

Results

The exercise group was significantly quicker than the control group in the timed up and go test (MD, −0.71; 95% CI [−1.09–0.32]; p < 0.001). There were also significant improvements in functional reach and a reduced fear of falling (MD, 4.34; 95% CI [1.39–7.30]; p = 0.005 and MD, −9.85; 95% CI [−0.19–−0.08]; p = 0.048, respectively).

Discussion

The improved TUG observed herein was better than the smallest real difference. Based on the results on mobility skills, balance and fear of falling, exergames may be an effective tool as a therapy for women with fibromyalgia.

Introduction

Fibromyalgia (FM) is a chronic syndrome characterized by widespread pain and other symptoms such as fatigue, trouble sleeping, anxiety, depression, impaired balance, stiffness, a high risk of falling, and poor physical fitness (Walker, 2016; Wolfe et al., 2010), all of which have a significant impact on quality of life (Huijnen et al., 2014). The estimated overall prevalence of FM in the general European population ranges from 2.9% to 4.7% (Branco et al., 2010), while it is lower in the United States (about 1.75%; 95% CI [1.42–2.07]) (Walitt et al., 2015). FM mostly affects women, who often experience higher levels of pain. On the other hand, men show higher disability and reduced physical function (Castro-Sanchez et al., 2012). Furthermore, men and women show different responses to therapy (Lami et al., 2016).

Treatment of fibromyalgia typically involves a combination of pharmacological and non-pharmacological approaches. According to Walker (2016) the main objectives are to improve the ability to perform activities of daily living by increasing physical function and reducing pain levels, which often lead to an improvement in quality of life. Exercise is usually the first step of a multidisciplinary approach (Walker, 2016). The effects of exercise on physical function (including balance and mobility skills) of women with FM are well documented. However, the effect size ranges from small to large depending on the type of exercise and the outcome measure used (Bidonde et al., 2014b). Although therapies based on whole body vibration (Adsuar et al., 2012), aquatic training (Tomas-Carus et al., 2009), or balance exercise training (Duruturk, Tuzun & Culhaoglu, 2015) effectively improve static and dynamic balance in women with FM, adherence to these exercise-based interventions is frequently poor (Oliver & Cronan, 2002); therefore, more motivating and enjoyable alternatives are required (Mortensen et al., 2015).

Exergames are a form of rehabilitation that combine the characteristics of physical exercise and the benefits of non-immersive virtual reality (VR) (Skip Rizzo et al., 2011), such as distraction from pain or enjoyment (Mortensen et al., 2015). This therapeutic modality has been used in older adults to improve fear of falling and balance during quiet standing under eyes open and eyes closed conditions (Gschwind et al., 2015; Van Diest et al., 2016) Exergames may be a motivating and enjoyable alternative form of rehabilitation (Burke et al., 2009; Herrero et al., 2014; Mortensen et al., 2015). However, to the best of our knowledge, no randomized controlled trial has examined the effects of exergames on patients with FM.

Therefore, the objective of this randomized controlled trial was to evaluate the effects of exergame-based intervention on mobility skills, balance and fear of falling in women with fibromyalgia.

Materials & Methods

Design

This randomized controlled trial was registered in the Australian New Zealand Clinical Trials Registry, ACTRN12615000836538. Two groups were included: an exercise group (EG) and a control group (CG). All patients were provided with written informed consent to participate and the study protocol was approved by the Committee of Bioethics of the University of Extremadura in May 2015 (reference number 61/2015). Randomisation and group allocation were performed by a single researcher. Allocation was concealed. Randomisation was based on computer-generated random code numbers. Half of the numbers were randomly assigned to the CG and the rest were assigned to the EG. The researchers in charge of evaluation were blind to this process. Participants were not blinded as they had been informed about the protocol and knew whether or not they were performing physical exercise.

Participants

The sample size was calculated to assess the effects of the intervention on impact of FM. This variable is often assessed using the FM impact questionnaire (FIQ) (Bennett, 2005). The minimum clinically important reduction in the FIQ was set at 14% of the total FIQ score, as described previously (Bennett et al., 2009a). For calculation purposes, the estimated mean FIQ was 70.5 ± 11.8, which was based on data from a previous study of a Spanish cohort (Esteve-Vives et al., 2007). Initially, the minimum power for detecting between-group differences of at least 14% was set at 85%, with an α value of 0.05. Based on these parameters, a minimum of 26 participants per group was necessary. No maximum number was set. A total of 83 women with FM fulfilled the inclusion criteria and were randomly assigned to the two groups. A maximum loss to follow-up of 15% was assumed. With respect to the main outcome measure (i.e., the TUG), the mean, standard deviation (SD), and smallest real difference were extracted from a previous study based on Spanish women with FM (Collado-Mateo et al., 2016). Assuming a mean (SD) of 6.791 (0.830), a difference of at least 9.33% (smallest real difference), with a power of 88% and an α value of 0.05 based on two groups (EG and CG), could be detected using a sample size of 35 participants per group.

Recruitment started in November 2015 and final measurements were carried out in May 2016. Recruitment was performed at two local FM associations. The inclusion criteria were as follows: (a) female, (b) aged 30–75 years, (c) diagnosed with FM by a rheumatologist according to the criteria of the American College of Rheumatology (Wolfe et al., 2010; Wolfe et al., 1990), (d) able to communicate effectively with the study staff, and (e) had read, understood, and signed the written informed consent form. Participants were excluded if they: (a) were pregnant, (b) changed their usual care therapies during the eight weeks of treatment, and, (c) had contraindications for physical exercise. Participants could abandon the study if (a) they withdrew informed consent, (b) the researcher or their general practitioner felt that they should withdraw from the study for reasons of safety, or (c) the participant did not attend at least 75% of the sessions.

Intervention

The intervention comprised eight weeks of VR-based physical exercise. The EG performed this activity twice a week for 1 h per session, whereas the CG received no treatment and continued their usual activities. All sessions were performed at the local association’s facilities. To increase the motivational and social component, exercises were performed in groups of three, with each person interacting with their own device. Intervention was based on an exergame called VirtualEx-FM, which was designed by the research group to improve physical conditioning of women with FM and increase their ability to perform activities of daily living. This exergame is based on Microsoft Kinect®, which connects to a computer and serves as the input sensor for tracking the movements of the participant. The software is run in the computer and contains a control panel through which the targets and difficulty can be modified. The system provides immediate visual feedback about the extent their movements fit the required patterns.

VirtualEx-FM provides three different virtual environments:

(a) The first comprises a warm-up followed by the aerobic part of the session. The participant has to imitate the movements that appear on the screen, which are performed by a professional kinesiologist and dance teacher, recorded, and stored in a software directory. Files can be removed, added, and modified as needed. Additionally, the playback speed can be manually controlled (0.5 ×, 1 ×, 1.5 ×, and 2 ×). In general, there were two types of video: warm-up videos based on full body joint movements, and dancing videos. The latter were based on Zumba, which is a fusion of aerobics and salsa.

(b) The second environment focuses on postural control and coordination. Participants interact with an apple that appears and disappears around them. The position at which the apple appears can be manually programmed by the technician. The sequence of which parts of the body interact with the apple can also be controlled and modified.

(c) The aim of the third environment is to improve mobility skills, balance and coordination. Participants are asked to step on virtual footprints. The distance between the footprints is controlled by the technician. The interface allows the technician to choose different types of steps: normal, tiptoe, heel-walking, raising the knees, and raising the heels.

The preferences (Lewis & Rosie, 2012) and needs (Bidonde et al., 2014a) of women with FM were considered when training in the three environments. Furthermore, VirtualEx-FM fulfilled the eight key points proposed by Lewis & Rosie (2012) for enhancing the efficacy of VR-based systems to optimize the rehabilitation benefits. These key points include all aspects related to the interface, social interaction, therapeutically principled movement patterns, progressively challenging games, and immediate feedback.

Outcome variables

All evaluations were performed in a laboratory at the university. Baseline data included age, years since FM diagnosis, BMI, and impact of FM as assessed by the revised version of the FM impact questionnaire (FIQ-R) (Bennett et al., 2009b).

Primary outcome

The timed up and go (TUG) (Podsiadlo & Richardson, 1991) test was used to assess mobility skills. The time taken by participants to rise from a chair, walk 3 m, turn around, walk back to the chair, and sit down again was measured. The best of three repetitions was recorded.

Secondary outcomes

Other variables were balance and fear of falling. Balance was evaluated using the functional reach (FR) and the Clinical Test of Sensory Integration of Balance (CTSIB) tests (Di Fabio & Seay, 1997). Fear of falling was assessed on a visual analogue scale, ranging from 0 to 100, where 0 was “no fear” and 100 was “extreme fear” (Scheffer et al., 2010).

The CTSIB is a widely used balance protocol and was conducted using the Biodex Balance System (Shirley, NY, USA). The test comprises four conditions: eyes open on a firm surface, eyes closed on a firm surface, eyes open on an unstable surface, and eyes closed on an unstable surface. Participants have to maintain their feet on the platform during the 30 s test, with a rest of 10 s between the different conditions. Higher sway indices would mean poorer balance. The FR (Duncan et al., 1990) measures the maximum distance participants can reach forward beyond arms’ length while keeping their feet on the floor, hip width apart. Two repetitions were performed and the best of them was analysed.

Data analysis

The mean effect (SD) for each group was calculated by subtracting the pre-intervention scores from the post-intervention scores. Between-group changes were calculated by subtracting the effects reported for the CG from the effects observed for the EG. A repeated-measures linear mixed model was used to compare the effects of the intervention between groups. Effect size was considered as low (<0.2), medium (>0.2 and <0.8) or large (>0.8) (Cohen, 1988).

Each group was divided into two subgroups to identify differences between women with and without fear of falling. For that purpose, the cut-off point was set at 50 because a previous study reported a mean of 48.88 in the VAS for women with fibromyalgia (Collado-Mateo et al., 2015).

Results

The flow of participants is shown in Fig. 1. In total, 86 women were screened and three were excluded for not meeting the inclusion criteria. These three women belonged to the association and suffered from fatigue and pain, but they were not diagnosed with FM by a rheumatologist. Therefore, 83 women were randomized into the EG (n = 42) and CG (n = 41). Table 1 summarizes the main baseline characteristics of both groups.

Figure 1 Flow of participants.

Table 1 Baseline characteristics of the study participants.

Variable (mean, SD)	EG (n = 41)	CG (n = 35)	
Age (years)	52.43 (9.83)	52.58 (9.42)	
Age distribution			
≤40 (frequency and %)	4 (9.8%)	3 (8.6%)	
41–60 (frequency and %)	28 (68.3%)	24 (68.6%)	
≥60 (frequency and %)	9 (22%)	8 (22.9%)	
Years since diagnosis (years)	10.36 (7.30)	12.48 (5.63)	
BMI (kg/m 2)	25.79 (5.14)	27.75 (5.62)	
Waist:hip ratio	0.85 (0.07)	0.86 (0.06)	
FIQ-R	46.21 (16.49)	43.64 (18.18)	
Notes.

EG Exercise group

CG Control group

TUG Timed-Up-Go

EOFS eyes open on a firm surface

ECFS eyes closed on a firm surface

EOUS eyes open on an unstable surface

ECUS eyes closed on an unstable surface

BMI Body Mass Index

Regarding the compliance with the treatment, forty-one of the forty-two women allocated to EG completed the intervention, which means that 2.38% of women from this group were lost to follow up. The cause of this loss was a change of residence. In the CG, 35/41 women attended the post-intervention session, meaning a loss of 14.63%. Intervention was considered complete when participants attended a minimum of 75% of sessions. Only one of the researchers was aware of the group to which each participant belonged. This researcher did not conduct the final measurements. No adverse effects were noted.

Table 2 compares the groups in terms of primary and secondary outcomes. For the TUG, the time taken for the EG to complete the test at the end of the study was 0.49 s (SD, 0.63) quicker than that at baseline. The time for the CG actually increased slightly (from 6.71 s at baseline to 6.93 s after the intervention). Therefore, the effect of exergames on the primary outcome was estimated as −0.71 (95% CI [−1.09–0.32]), meaning a significant improvement of 10.61% (p < 0.001).

Table 2 Effects of the 8-week training program on the VirtualEx-FM exergame and control groups (n = 76).

	EG (n = 41)	CG (n = 35)	Differences between interventions [mean (95% CI)]*	F-value	P-valuea	P-value (age as covariate)	Effect size	
	Pre- intervention, mean (SD)	Post- intervention, mean (SD)	Pre- intervention, mean (SD)	Post- intervention, mean (SD)						
TUG (s)	6.69 (0.90)	6.19 (0.61)	6.71 (0.78)	6.93 (1.13)	−0.71 (−1.09–0.32)	13.692	<0.001	<0.001	0.86	
Balance EOFS (°)	0.53 (0.20)	0.58 (0.22)	0.67 (0.34)	0.67 (0.53)	0.05 (−0.08–0.19)	0.637	0.427	0.432	0.19	
Balance ECFS (°)	0.99 (0.53)	0.85 (0.51)	0.98 (0.67)	1.08 (0.95)	−0.24 (−0.50–0.01)	3.480	0.066	0.067	0.41	
Balance EOUS (°)	1.18 (0.42)	1.00 (0.22)	1.23 (0.62)	1.21 (0.62)	−0.16 (−0.33–0.00)	3.802	0.055	0.056	0.31	
Balance ECUS (°)	3.25 (0.93)	2.76 (0.66)	2.79 (1.76)	2.73 (1.19)	−0.42 (−0.82–0.02)	4.555	0.036	0.037	0.32	
Functional reach (cm)	19.71 (7.54)	23.00 (5.15)	19.49 (5.70)	18.43 (6.62)	4.34 (1.39–7.30)	8.582	0.005	0.005	0.65	
Fear of falling	26.10 (28.53)	21.09 (25.92)	28.57 (30.40)	33.42 (32.17)	−9.85 (−0.19–0.08)	4.044	0.048	0.048	0.34	
Notes.

EG Exercise group

CG Control group

TUG Timed-Up-Go

EOFS eyes open on a firm surface

ECFS eyes closed on a firm surface

EOUS eyes open on an unstable surface

ECUS eyes closed on an unstable surface.

SD Standard deviation.

* Differences between interventions: calculated as (post-intervention EG—pre-intervention EG)—(post-intervention CG—pre-intervention CG).

a P-values calculated using analysis of variance (ANOVA) for repeated measures.

In the CTSIB protocol, the improvement was significantly higher for the EG only under the condition of eyes closed on an unstable surface (p = 0.036). Results from the ANOVA almost reached significance for the condition of eyes closed on a stable surface condition and of eyes open on an unstable surface (p = 0.066 and 0.055, respectively). Significant improvement was observed in the FR test (p = 0.005), the EG improved by 3.53 cm (an improvement of 23.58%), whereas the CG deteriorated by 1.06 cm.

Finally, the score for fear of falling in the EG fell from 26.10 to 21.09, while that in the CG increased from 28.57 to 33.42. Results from the ANOVA reached significance (p = 0.048). As shown in Table 3, the reduction was highest in participants with the greatest fear of falling. The most fearful in the EG reduced their score from 63.46 to 43.84, a reduction of 30%.

Table 3 Effects of VirtualEx-FM on fear of falling in the exergame and control groups (n = 76).

	Fearless EG (fear of falling < 50; n = 28)	Fearless CG (fear of falling < 50; n = 27)	Difference between intervention (virtual minus control)*	P-value interactiona	Effect size	
	Pre-intervention, mean (SD)	Post-intervention, mean (SD)	Pre-intervention, mean (SD)	Post-intervention, mean (SD)				
Fear of falling	8.75 (12.29)	10.53 (18.97)	14.44 (14.76)	21.85 (25.87)	−5.62 (−16.09–4.85)	0.287	0.28	
	Fearful EG (fear of falling > 50; n = 13)	Fearful CG (fear of falling > 50; n = 8)	Differences between interventions (virtual minus control)	P-value interaction	Effect size	
	Pre-intervention, mean (SD)	Post-intervention, mean (SD)	Pre-intervention, mean (SD)	Post-intervention, mean (SD)				
Fear of falling	63.46 (12.64)	43.84 (24.67)	76.25 (17.67)	72.50 (16.69)	−15.86 (−36.17–4.44)	0.119	1.13	
Notes.

EG Exercise group

CG Control group

SD Standard deviation

* Differences between interventions: calculated as (post-intervention EG—pre-intervention EG)—(post-intervention CG—pre-intervention CG).

a P-values calculated using analysis of variance (ANOVA) for repeated measures.

Discussion

This is the first randomized controlled trial to examine the utility of exergames as a therapy for improving mobility skills and balance in people with FM. These variables were improved through challenging exercises performed using a VR system. The improved TUG observed herein was slightly better than the smallest real difference reported in a previous study of women with FM (Collado-Mateo et al., 2016), which was reported as 9.33%. Therefore, the improvement in mobility skills can be considered relevant. The rest of the measures assessed in the present study focused on static balance, since both the CTSIB protocol and the FR test are performed without displacement of the feet. The baseline sway indices for each condition in the CTSIB protocol were better than those reported in a previous study of women with FM (Collado-Mateo et al., 2015); the exception was the condition “eyes closed on an unstable surface”. In the present study, this condition was significantly improved from 3.25°at baseline (note that in the previous study, the mean sway index for women aged >60 was 3.29°) to 2.76°(in the previous study, the mean value for women with FM aged <50 was 2.77°). Therefore, balance performance under the condition of standing with eyes closed on an unstable surface changed from 3.25°(close to the mean for patients aged >60) to 2.76°(close to the mean for patients aged <50). The improvement in the FR test was also significant, with a between-group change of more than 23%.

Physical inactivity and a sedentary lifestyle are common in FM patients, which may be aggravated by a greater fear of falling. This is especially relevant in this population, where around 72% are overweight (Segura-Jimenez et al., 2015). By reducing the fear of falling, women with FM may be able to increase their participation in physical, social, and other types of activity, thereby improving their quality of life.

The impact of fear of falling on the activities of daily living of women with FM was examined by Rutledge et al. (2013): persons with FM often reported frustration because they want to continue normal activities but are not able due to fear of falling. The reduction of fear of falling in the EG was almost 20%, which may be interpreted as a great improvement. In the present study, fear of falling at baseline was lower than that reported in a previous study (Collado-Mateo et al., 2015) in which the score ranged from 37 to 55. To explore this improvement more deeply, the cohort was divided into two groups: women that scored ≥50 on the fear of falling scale and those that scored <50. Those in the EG with the greatest fear reduced their score from 63.46 to 43.84, an improvement of almost 20 points. On the other hand, the score for the group with the least fear increased from 8.75 to 10.53. This was not observed for the CG, in which the improvement in the higher fear group was less than 5%, while the score for the low fear group increased by 7.4 points. Therefore, exergames have a large effect on fear of falling in women that already experience higher levels of fear. By contrast, they have little effect on those with low levels of fear.

The effect of age on balance and mobility skills is well documented. The present study included women aged between 30 and 71, which is a wide range. Although the mean (SD) and distribution of age was extremely similar in EG and CG, further attempts to evaluate the role of age on the treatment effects were performed. First, “age” was included as a covariate in the model, which led to only minimal changes in the p-values (see Table 2). Second, a correlation analysis of the relationship between the treatment effects and age was performed. No statistically significant correlation was observed. Therefore, the effects of exergame on mobility skills and balance seem not to have been influenced by age in the current study. However, future studies should include different age groups or restrict the age range.

Two main limitations should be mentioned. First, only women were included in the study due to potential gender differences (Castro-Sanchez et al., 2012; Lami et al., 2016). Therefore, the effects and applicability of exergame intervention on men with FM remain unknown. Second, the study lacked an active CG. However, given the well-known effects of different types of exercise on FM, effect sizes can be compared with those reported in the scientific literature.

Conclusions

Exergames are an effective tool for improving mobility skills, balance and fear of falling in women with FM. The improvement in the TUG was better than the smallest real difference. Regarding balance, significant improvements were observed in FR and in the CTSIB with eyes closed on an unstable surface. These results, along with high adherence (only one participant abandoned the intervention), indicate that exergames may be a feasible alternative form of rehabilitation therapy for improving balance and mobility problems in this population.

Supplemental Information

Supplemental Information 1 Completed CONSORT checklist

Click here for additional data file.

Data S1 Raw data

Click here for additional data file.

Additional Information and Declarations

Competing Interests

Author Contributions

Human Ethics

Clinical Trial Ethics

Data Availability

Clinical Trial Registration

The authors declare there are no competing interests.

Daniel Collado-Mateo and Jose C. Adsuar conceived and designed the experiments, performed the experiments, analyzed the data, contributed reagents/materials/analysis tools, wrote the paper, reviewed drafts of the paper.

Francisco J. Dominguez-Muñoz conceived and designed the experiments, performed the experiments, analyzed the data, contributed reagents/materials/analysis tools, prepared figures and/or tables, reviewed drafts of the paper.

Eugenio Merellano-Navarro performed the experiments, analyzed the data, contributed reagents/materials/analysis tools, wrote the paper, prepared figures and/or tables, reviewed drafts of the paper.

Narcis Gusi conceived and designed the experiments, analyzed the data, contributed reagents/materials/analysis tools, reviewed drafts of the paper.

The following information was supplied relating to ethical approvals (i.e., approving body and any reference numbers):

The University of Extremadura granted Ethical approval to carry out the study. Reference number 61/2015.

The following information was supplied relating to ethical approvals (i.e., approving body and any reference numbers):

The University of Extremadura granted Ethical approval to carry out the study. Reference number 61/2015.

The following information was supplied regarding data availability:

The raw data has been supplied as a Supplementary File.

The following information was supplied regarding Clinical Trial registration:

Australian New Zealand Clinical Trials Registry, ACTRN12615000836538.

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
