# Peer review of "Exergames for women with fibromyalgia: a randomised controlled trial to evaluate the effects on mobility skills, balance and fear of falling"

_PeerJ, doi:10.7717/peerj.3211_

## Round 0.1 · original submission · Major Revisions

· Academic Editor

Major Revisions

Dear authors:

Your manuscript was evaluated by expert reviewers.

Please address reviewer´s concerns in detail.

Also, take this chance to:

-improve the writing style and clarity of your paper.
-adapt your manuscript to PeerJ policies.
-use clear and unambiguous text that conforms to professional standards.
-include sufficient introduction and background to demonstrate how the work fits into the broader field of knowledge, with relevant prior literature appropriately referenced.
-use the format of ‘standard sections’ from PeerJ Instructions for Authors (a significant modification from PeerJ suggested structure should be made only if this modification significantly improve clarity or conform to a discipline-specific custom).
-your research question must be clearly defined, must be relevant and meaningful, identifying the knowledge gap being investigated and how the study contributes to filling that gap.
-provide a Methods section with sufficient information to be reproducible by another investigator.
-review that your data be robust and statistically sound.
-assure that your conclusions be appropriately stated, connected to the original question investigated, and limited to those supported by the results.
-speculation (especially in the discussion of your results) is welcomed, but should be identified as such.


Sincerely,
Rodrigo Ramírez-Campillo
Academic Editor
PeerJ

Reviewer 1 ·

Basic reporting

- The article has been written in English in a clear and precise way.
- The introduction demonstrates sufficient background to understand that balance and fear of falling are one of the major problems in female patients with fibromyalgia. The authors suggest that exergame therapy is a novel intervention that may improve previous variables because of the ability to motivate patients to continue and perpetuate therapy. However, the introduction does not show a clear hypothesis in favor of this therapy.
- On the other hand, the antecedents of Barry et al (2016) allows to infer that exergame intervention is a potentially beneficial therapy to increase adherence to the treatment of patients (without defining the pathology). This problem is common in rehabilitation, however, the objective of the research does not contemplate this problem and focuses only on the possible effects of exergame. Because the research did not consider assessing adherence to treatment, I suggest deleting the information from Barry et al (2016) or performing an inferential analysis of the data obtained.
- L38: Replace "postural stability" by a more representative balance key word, such as "postural balance" or "postural control". In addition, replace "pain" by "chronic pain", because fibromyalgia is considered as a pathology that experiences chronic pain.
- L43-44: The sentence is redundant: "all of which have a significant impact on quality of life and the ability to perform activities of daily living". I suggest deleting "and the ability to perform daily living activities".
- L56-58: Affirmation needs references.
- L59-63: The writing is unclear and contradictory. The sentence "However, evidence suggests that exercise interventions (aquatic resistance, or a combination of these) improve physical function;". The idea is more clear if it is written this way "The effects of exercise on the physical function (including balance) of women with FM are well documented. However, the effect size ranges from small to large depending on the type of exercise and the measure used (Bidonde, Busch, Webber, et al., 2014).
- The manuscript is structured according to PeerJ standards, has a well organized abstract of 243 words. The manuscript includes all the results of the research.
L84-94: The paragraph does not have the same line spacing of the entire manuscript (1,5).
Table 1: The legend has abbreviations that are not used. Check.
Table 2: The legend has a super index "a" which indicates a calculation of the p-value through a t-test for paired samples. However, the table does not display the super index. The same case occurs with the super index "b".
- Table 3: The table is visually confusing. Improve their representation. The same problem occurs in relation to the super indices in Table 2.
- Figure 1: In the "allocation" box, design the sentence "Did not receive allocated intervention (n = 0)" in a single line.

Experimental design

The manuscript complies with PeerJ's Aims and Scope.
- The research question is well defined. Only the information of Barry et al (2016) confuses what the research problem is.
- The research complies with rigorous methodological and ethical criteria of the international research community, plus it includes a registry of clinical trials: "This randomized controlled trial was registered in the Australian New Zealand Clinical Trials Registry, ACTRN12615000836538. Two groups were included: an exercise group (EG) and a control group (CG). All patients were provided with written informed consent to participate and the study protocol was approved by the local ethics committee in May 2015 (reference number 89 61/2015). "
- The research has an inclusion criterion that can increase the probability of type 1 error. The criterion "b) aged 30-75 years" is a very broad age range that can influence the results of the postural balance, since several authors have shown that there is a close relationship between the age and the physiological maturity of the sensorial components (Hirabayashi & Iwasaki, 1995, Hatzitaki et al., 2002, Steindl et al., 2006, Gatica et al., 2013, Gatica et al., 2014). Older adults have deterioration of the visual system, which can influence the results. Therefore, it is suggested to include a criterion "age 30-60 years" and perform a new statistical analysis. Another option is to balance the groups according to the ages.
- The research does not indicate the place of evaluation nor the environmental conditions of evaluation. This makes the reproducibility of research difficult.
- L171: The CTSIB does not indicate the units of measurement.
- L185-186: What is the criterion and your reference for deciding a cut-off score of 50?
- Statistical analysis does not indicate post-hoc analysis. Include classification of effect size values.
- The results of ANOVA should indicate the value F, p and gl. Include in all the indicated results.
- The value p = 0.055 is not a significant value since it is greater than 0.050.

Validity of the findings

- The results are novel from the point of view of the contribution to the rehabilitation of patients with chronic pain, who have few strategies to reduce the consequences of the disease. However, the results should be taken with caution, since the research did not consider a control that included conventional therapy (aerobic exercise, stretching, alternative therapies, etc.).
- The results may be influenced by the inclusion criteria. Considering a range from 30 to 75 years may increase the likelihood of committing a type 1 error. In addition, the study does not have a baseline characterization table of subjects that reveals the ages of each subject or age range (eg 30 to 40, 41 to 50, 51 to 60, etc.) as well as a statistic that analyzes the primary and secondary variables by age ranges. Taking these measures can improve the validity of the results.
- The conclusion is related to the objective of the investigation.

Additional comments

- The research is novel and original as it tries to solve a frequent research problem in patients with chronic pain. However, there are drawbacks in the eligibility criteria that make it difficult to validate the results (wide age range "30 to 75 years"). It is suggested to perform an analysis by age ranges or not to consider adults 60 years or older according to evidence from the literature (Hirabayashi & Iwasaki, 1995; Hatzitaki et al., 2002; Gatica et al., 2013; Gatica et al., 2014).
I suggest asking the researchers a substantial modification of the methodology of the research in relation to the inclusion criteria and its possible modification (see comments above). If these substantial changes are made, the research will have greater internal robustness and thus its results can be compared in future investigations. If these suggestions are not applied, the results will have no validity and there will be a high probability of committing a type 1 error.

·

Basic reporting

All comments in pdf

Experimental design

All comments in pdf

Validity of the findings

All comments in pdf

Additional comments

All comments in pdf

---

## Round 0.2 · Minor Revisions

· Academic Editor

Minor Revisions

Dear Daniel,

the reviewers (and myself) are broadly satisfied with the sophistication of the manuscript. However, some minor revisions are still required (please, see document linked below).

Reviewer 1 ·

Basic reporting

No comment.

Experimental design

No comment.

Validity of the findings

No comment.

Additional comments

The authors made the proposed changes and markedly improved the writing of the manuscript. I recommend the publication of this article.

---

## Round 0.3 · accepted · Accept

· Academic Editor

Accept

I am pleased to inform you of the official acceptance of your manuscript for publication in PeerJ.

Thank you very much for the opportunity to review your manuscript and congratulations.

·

Basic reporting

No further comments/suggestions

Experimental design

No further comments/suggestions

Validity of the findings

No further comments/suggestions

Additional comments

Dear Authors:

All my concerns in regards to the last revision have been answered and I am content with them. I do really appreciate the effort in doing this RCT as well as dealing with my comments and suggestions.

Regards